# The Choice of Anti-Inflammatory Influences the Elimination of Protein-Bound Uremic Toxins

**DOI:** 10.3390/toxins16120545

**Published:** 2024-12-16

**Authors:** Víctor Joaquín Escudero-Saiz, Elena Cuadrado-Payán, María Rodriguez-Garcia, Gregori Casals, Lida María Rodas, Néstor Fontseré, María del Carmen Salgado, Carla Bastida, Nayra Rico, José Jesús Broseta, Francisco Maduell

**Affiliations:** 1Nephrology and Renal Transplantation, Hospital Clínic de Barcelona, 08036 Barcelona, Spain; vjescudero@clinic.cat (V.J.E.-S.); ecuadrado@clinic.cat (E.C.-P.); lmrodas@clinic.cat (L.M.R.); fontsere@clinic.cat (N.F.); jjbroseta@clinic.cat (J.J.B.); 2Biochemistry and Molecular Genetics Department-CDB, Hospital Clínic de Barcelona, 08036 Barcelona, Spain; mrodriguezg@clinic.cat (M.R.-G.); casals@clinic.cat (G.C.); salgado@clinic.cat (M.d.C.S.); nrico@clinic.cat (N.R.); 3Pharmacy Department, University of Barcelona, 08036 Barcelona, Spain; cbastida@clinic.cat; 4Medicine Department, University of Barcelona, 08036 Barcelona, Spain

**Keywords:** hemodiafiltration, uremic toxins, dialysis efficiency, pain, protein-bound toxins, p-cresyl sulfate, indoxyl sulfate

## Abstract

Pain is a frequent and disturbing symptom among hemodialysis patients. Protein-bound uremic toxins (PBUTs) are related to cardiovascular and overall mortality, and they are difficult to remove with current hemodialysis treatments. The PBUT displacers, such as furosemide, tryptophan, or ibuprofen, may be promising new strategies for improving their clearance. This study aims to compare ibuprofen versus other analgesic drugs in PBUT removal. A prospective study was carried out in 23 patients. Patients underwent four dialysis sessions with routine dialysis parameters, except for analgesic drugs administered (lysine acetylsalicylic acid, acetaminophen, dexketoprofen, and ibuprofen). The reduction ratios (RRs) of a wide range of molecular weight molecules were assessed, including total p-cresyl sulfate and total indoxyl-sulfate. There were no complications related to the administered drug, and pain was controlled independently of the drug. There were no differences in the RR of small-size and medium-sized molecules between all four study treatments. However, indoxyl sulfate and p-cresyl sulfate RRs when ibuprofen was administered were significantly higher than lysine acetylsalicylic acid, acetaminophen, and dexketoprofen treatments. In conclusion, patients with pain may benefit from treatment with ibuprofen instead of lysine acetylsalicylic acid, paracetamol, or dexketoprofen, since in addition to improving pain, it increases the removal of PBUTs.

## 1. Introduction

Pain is a common and multifactorial symptom experienced by patients undergoing hemodialysis, affecting nearly 60% of this population, and it is consistently associated with a lower health-related quality of life [1,2]. Currently, there are various strategies, both non-pharmacological and pharmacological, to manage pain. Among the analgesic medications frequently used are acetaminophen and non-steroidal anti-inflammatory drugs (NSAIDs) [3].

The European Uremic Toxins (EUTox) working group classified uremic toxins (UTs) into three groups depending on their physicochemical properties, which influence their clearance in conventional hemodialysis [4]. Protein-bound uremic toxins (PBUTs) are low-molecular-weight UTs but have a high affinity for plasma proteins, which limits their clearance by high-flux hemodialysis, post-dilutional hemodiafiltration (HDF), or extended and expanded hemodialysis [5,6,7,8]. PBUTs encompass more than 100 molecules [9], with indoxyl sulfate (IS) and p-cresyl sulfate (pCS) being two of the most toxic [10]. IS and pCS are related to increased cardiovascular disease and mortality among dialysis patients [11,12].

Some strategies for enhancing PBUT clearance are under development [13]. The use of PBUT displacers, molecules that compete for their albumin-binding site, has been used in vitro with satisfactory results [14]. Some examples include tryptophan, furosemide, ibuprofen, or free fatty acids [13]. However, among all those tested to date, ibuprofen is the only one studied in vivo with increased PBUT clearance in hemodialysis patients [15].

Since, in our unit, 25% of patients need intradialytic non-opioid analgesia, this study aimed to compare the influence of ibuprofen versus lysine acetylsalicylic acid, acetaminophen, and dexketoprofen analgesic drugs on the removal of a wide range of molecular weight molecules, specifically including total pCS and IS as markers of PBUTs.

## 2. Results

All 23 included patients completed all sessions of the protocol without drug adverse reactions or notable clinical incidents. There were no differences in Qb, real duration, total blood processed, vascular access recirculation, initial weight, final weight, weight gain, initial and final hematocrit, arterial pressure, venous pressure, TMP, and convective volume (Table 1).

### 2.1. Small-Sized Molecules

The dialysis diffusive dose, measured by urea and creatinine RRs, showed that there were no differences in all four treatments (Table 2).

### 2.2. Medium-Sized Molecules

There were no differences in β_2_-microglobulin, myoglobin, free kFLC, prolactin, α_1_-microglobulin, α_1_-acid glycoprotein, and free λFLC RRs (Table 1). There were also no differences in albumin RRs (Table 2).

### 2.3. Protein Bound Uremic Toxins

Average values of indoxyl sulfate RRs ranged between 40% and 70%. Average values of p-cresyl sulfate RRs ranged between 40% and 65%. There were no differences in indoxyl sulfate and p-cresyl sulfate RRs between AAS, paracetamol, and dexketoprofen. However, indoxyl sulfate and p-cresyl sulfate RRs when ibuprofen was administrated were significantly higher than in the other three study situations (Figure 1). Individual values are shown in Appendix A.

### 2.4. Global Removal Score

There were no differences in GRS between all treatments evaluated (Figure 2).

## 3. Discussion

The results of the present study, which compared the effects on the clearance of various molecular weight molecules, show that ibuprofen does not differ from lysine acetylsalicylic acid, dexketoprofen, and acetaminophen in post-dilutional HDF patients. Notably, the only exceptions are the PBUTs, both IS and pCS, as ibuprofen significantly influences their displacement.

Due to the high protein binding of pCS and IS, removing these substances is challenging, regardless of the dialysis modality used. Published studies indicate that low-flux hemodialysis achieves an RR of IS of 22% with a Qb of 200 mL/min and a Qd of 300 mL/min [16]. Lessafler et al. reported RRs of 32.9% for pCS and 42.4% for IS when using Qb of 250 mL/min and Qd of 500 mL/min over a 4 h session [17]. In high-flux hemodialysis, studies show RRs for pCS and IS during a 4 h session ranging from 27% to 47% and 33% to 52%, respectively [6,7,8,17,18,19]. For an 8 h duration, RRs of 37% [20] and 45% [7] for pCS, as well as 43% [20] and 55% [7] for IS, were achieved. Regarding pre-dilutional HDF, Meert et al. report 41–45% RRs for pCS and 48% for IS [21,22]. In 4 h of post-dilutional HDF, studies show RRs for pCS ranging from 38% to 48% and for IS between 45% and 55% [6,7,8,19,21,22,23,24]; while for an 8 h post-dilutional HDF, the RR for pCS was 52%, and for IS was 60% [7]. Only one study on mid-dilutional HDF reported an RR of 42.7% for pCS and 47.4% for IS with Qb of 343 mL/min and Qd of 650 mL/min over 250 min [23]. In a 4 h HDF session with endogenous reinfusion—utilizing diffusion, convection, and adsorption—Esquivias-Motta et al. [19] achieved RRs of 50.7% for pCS and 48.8% for IS with Qb exceeding 350 mL/min; and Chen et al. found RRs of 40.9% for pCS and 43.6% for IS with Qb of 249 mL/min [25]. There is also one study with expanded hemodialysis that, with Qb of 300 mL/min and Qd of 530 mL/min, reported RRs of 29.5% for pCS and 36.3% for IS [6]. Our study yielded results consistent with this existing literature, as it was conducted over 5 h with a Qb of 426 mL/min using post-dilutional HDF; we achieved a higher RR for IS and a similar one for pCS without ibuprofen compared to other works with post-dilutional HDF; moreover, these figures increased when using ibuprofen. The clearance of PBUTs is primarily diffusive, as these smaller molecules depend on blood and dialysate flows. Thus, extended dialysis modalities may be more effective, as they facilitate the dissociation of bound toxins while allowing PBUTs to refill from extravascular compartments. It is still necessary to demonstrate the further role of adsorption added to diffusion and convection in the clearance of PBUTs.

Nowadays, there is growing interest in improving dialysis treatment for enhancing the clearance of PBUTs. In this context, the use of displacers or competitive agents is at the forefront with current in vitro and in vivo experiences, but there is still a long way to progress in which the development of new displacers is an interesting option. IS and pCS bind primarily to Sudlow site II of albumin [26,27,28,29]. This finding is supported by structural analyses highlighting their high-affinity binding patterns to this site, limiting their availability for clearance during hemodialysis [28,29] (Figure 3). Some studies have shown the displacer capability of ibuprofen in vitro, which also binds to Sudlow site II and increases the free fraction of these PBUTs [30,31]. Other non-steroidal anti-inflammatory drugs also bind to albumin, such as lysine acid acetylsalicylic, dexketoprofen, and acetaminophen [32], but they do not compete with IS and pCS for the same binding site, which may explain the similarity in their RRs in our work and the superiority of ibuprofen. Madero et al. were the first to demonstrate ibuprofen’s efficacy in vivo in enhancing IS and pCS sulfate clearances during its infusion in 4 h of high-flux hemodialysis from 6 to 20.2 mL/min and from 4.4 to 14.9 mL/min, respectively [15]. Even when these results refer to the clearance of these PBUTs, the results of our study using a 5 h post-dilutional HDF modality indicate a 12.9% increase in IS and a 14.2% increase in pCS RRs, which would be consistent with their findings.

Other displacers, such as furosemide or tryptophan, increase IS and pCS free-fraction in vitro, but no available data exists in vivo [30,31]. Ibuprofen also showed higher binding affinity to albumin compared to IS, pCS, and also to these other displacers [27,31]. Some other ones, such as salvianolic acids [34] or intravenous lipid emulsions (Intralipid^TM^, Fresenius KABI SSPC, Jiangsu, China) [35], have been tested in rat models with positive results, while Intralipid^TM^ is the first to be effective against other PBUTs like indole-3-acetic acid (IAA) and 3-carboxy-4-methyl-5-propyl-2-furanpropionic acid (CMPF). Interestingly, salvianolic acids bind to Sudlow site I while IS and pCS bind predominantly to Sudlow site II [26,27,28,29], so their displacement ability is due to an allosteric mechanism instead of direct competition [35]. Moreover, free-fatty acids from Intralipid^TM^ have competitive displacement, as some have higher binding affinity to Sudlow site I, which coincides with fatty-acid (FA) binding site 7 (FA7), compared to CMPF. Others, such as oleic and linoleic acids, bind with higher affinity to Sudlow site II (FA3-FA4) compared to IS and pCS. These FA bindings on albumin may change the structure of the PBUT binding sites [35] (Figure 3). However, there is no current data about their security in humans, and further studies are needed to assess the potential side effects of long-term treatment with these displacers [36]. Another innovative field is the development of adsorptive membranes that capture mainly the protein-bound fraction while facing the difficulty of biocompatibility or material saturation. Moreover, this membrane evolution has led to the idea of competitive membranes in which a PBUT displacer is covalently bound to the polymer matrix of the membrane [36].

The use of non-steroidal anti-inflammatory drugs (NSAIDs) might be cautious because of their widely known side effects in the general population, such as increased risk of gastrointestinal bleeding, increased cardiovascular risk, or worsening residual kidney function. However, studies that focus on their long-term outcomes in the hemodialysis population are scarce, as other risk factors may be coexistent (e.g., antiplatelets and/or anti-thrombotic agents) [37]. We did not observe any drug-related complications during and after the treatment sessions during the study. All in mind, hemodialysis patients who receive NSAIDs must be closely monitored for potential side effects, but when no potential risk factors for them concur, ibuprofen may be chosen as the preferable analgesic treatment due to this added benefit. Nonetheless, further studies with more statistical potential are needed to define whether this effect is clinically relevant or to look for the most secure/safe patient profile.

This study faces some limitations. Firstly, we only measure total IS and pCS as the main PBUTs, but others, such as IAA or CMPF, have not been measured. Secondly, ibuprofen binds to Sudlow site II of albumin, but PBUTs bind to Sudlow site I as well, so it would be considered an incomplete or selective displacer for some PBUTs but not all. Third, if this increased removal is clinically relevant, randomized and masked clinical trials must demonstrate it.

## 4. Conclusions

Hemodialysis patients with pain may benefit from treatment with ibuprofen instead of lysine salicylic acid, dexketoprofen, or acetaminophen since it improves pain at the same time that it increases the removal of some PBUTs as IS and pCS. Nonetheless, further long-term studies are necessary to confirm these recommendations.

## 5. Materials and Methods

### 5.1. Study Design

This prospective, single-center study included 23 patients with a prescription of intradialytic treatment for pain, 12 men and 11 women; the mean age was 71 ± 14 years (range 38 to 91) on a regular hemodialysis program for 66 ± 81 months (range 11–310). Vascular accesses were autologous arteriovenous fistula (AVF) in 13 patients, tunneled catheter in 8, and prosthetic arteriovenous fistula in the other two patients. The anticoagulation used was low-molecular-weight heparin in 16 patients and heparin sodium in 6 patients; the remaining patient was dialyzed without heparin. Underlying renal diseases were nephroangiosclerosis (seven patients), diabetic nephropathy (six patients), chronic glomerulonephritis (four patients), urologic etiology (three patients), interstitial nephritis (one patient), and undiagnosed nephropathy (two patients). All patients provided informed consent. The study was approved by the local ethics committee and was conducted according to the principles of the Declaration of Helsinki.

Each patient underwent four dialysis sessions with routine dialysis parameters in which only the anti-inflammatory treatment was modified (Figure 4):-Lysine acetylsalicylic acid 900 mg.-Acetaminophen 1 g.-Dexketoprofen 50 mg.-Ibuprofen 800 mg.

The prescribed dialysis parameters were post-dilution HDF, dialysis buffer with bicarbonate, dialysate flow 400 mL/min, blood flow (Qb) 426 ± 26 mL/min (range 400–450), and dialysis time 295 ± 12 min (range 240–300). Each patient maintained the same dialyzer (14 FX 60 Cordiax, 7 Clearum, and 2 Solacea). Net fluid removal will be set individually, depending on the patient’s clinical needs. Fresenius 5008 Cordiax or 6008 CAREsystem (Fresenius, Bad Homburg, Germany) dialysis monitors were used. The variables collected in each session were as follows: real duration, dialyzer, Qb, recirculation index measured by the temperature module, arterial and venous pressure, transmembrane pressure (TMP), initial and final hematocrit automatically measured by the BVM^®^ biosensor (Fresenius, Bad Homburg, Germany) initial and final body weights, volume of blood processed, and replacement volume.

Blood and dialysis fluid samples for analyses were taken from each patient in the same dialysis session of the week. Laboratory measurements included concentrations of urea (molecular weight [MW] 60), creatinine (MW 113), ß_2_-microglobulin (MW 11,800), myoglobin (MW 17,200), kappa-free immunoglobulin light chains (κFLC, MW 22,500), prolactin (MW 23,000), α_1_-microglobulin (MW 33,000), α_1_-acid glycoprotein (MW 41,000), lambda-free immunoglobulin light chains (λFLC, MW 45,000), and albumin (MW 66,000) in serum at the beginning and at the end of each session to calculate the reduction ratio (RR) of these solutes. Protein-bound uremic toxins (PBUT), p-cresyl sulfate (MW 108) and indoxyl sulfate (MW 213), were also evaluated. The final concentrations of ß2-microglobulin, myoglobin, κFLC, prolactin, α_1_-microglobulin, α_1_-acid glycoprotein, λFLC, p-cresyl sulfate, indoxyl sulfate, and albumin were corrected for the degree of hemoconcentration and the volume of distribution (approximate extracellular volume) according to Bergström and Wehle [38].

Urea and creatinine were measured by molecular absorption spectrometry, albumin and β_2_-microglobulin were measured by inmunoturbidimetry, and myoglobin and prolactin were measured by indirect enzyme immunoassay (EIA); all of them were performed in the Atellica Solution analyzer (Siemens Healthineers, Tarrytown, NY, USA). Indoxyl sulfate and p-cresyl sulfate were measured in serum using liquid chromatography-mass spectrometry (LC-MS). Finally, α_1_-microglobuline, α_1_-acid glycoprotein, κFLC, and λFLC were measured by inmunonephelometry using the BNII analyzer (Siemens Healthineers, Erlangen, Germany). We used the global removal score (GRS) to evaluate the effectiveness of a removal dialyzer, including the RR of molecules with MW ranging from 60 to 41,000 and considering albumin RR as negative values, calculated with the formula [39]:$$\frac{{\mathrm{Urea}}_{\mathrm{RR}}+{\beta_{2}\text{-}\mathrm{microglobulin}}_{\mathrm{RR}}+{\mathrm{myoglobin}}_{\mathrm{RR}}+{\mathrm{prolactin}}_{\mathrm{RR}}+{\alpha_{1}\text{-}\mathrm{microglobulin}}_{\mathrm{RR}}+{\alpha_{1}\text{-}\mathrm{acid}\;\mathrm{glycoprotein}}_{\mathrm{RR}}\text{-}{\mathrm{albumin}}_{\mathrm{RR}}}{6}$$

### 5.2. Statistical Analysis

The results are expressed as the arithmetic mean ± standard deviation. For the analysis of the statistical significance of quantitative parameters, the Student *t* test for paired data and ANOVA for repeated data followed by Bonferroni’s post hoc comparisons tests were used for the parametric data. A *p* < 0.05 was considered statistically significant. Analyses were performed using SPSS software version 23 (SPSS, Chicago, IL, USA), and graphics were performed using Prism version 10 (GraphPad Software, Boston, MA, USA).

## Figures and Tables

**Figure 1 toxins-16-00545-f001:**
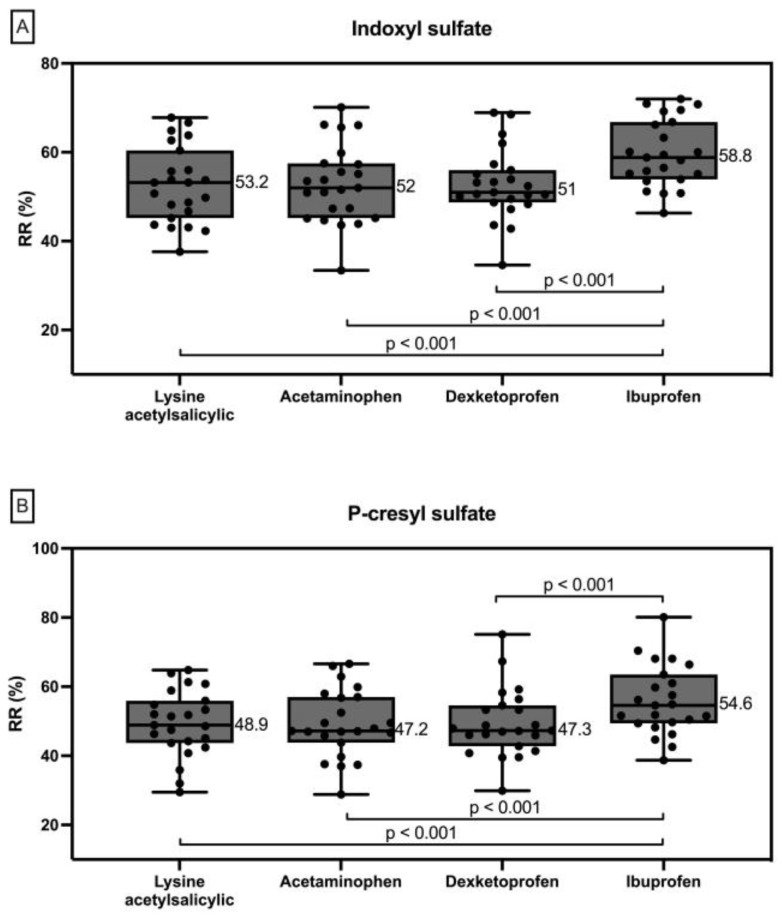
Indoxyl sulfate (**A**) and p-cresyl sulfate (**B**) RRs when lysine acetylsalicylic, acetaminophen, dexketoprofen and ibuprofen (left to right) are administered.

**Figure 2 toxins-16-00545-f002:**
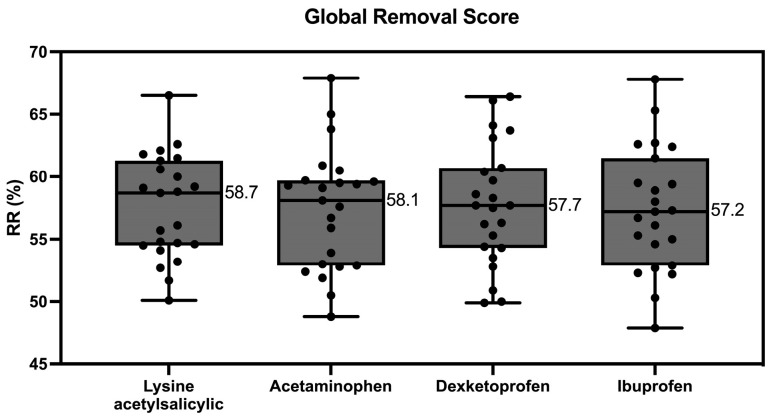
Global removal score when lysine acetylsalicylic, acetaminophen, dexketoprofen and ibuprofen are administered.

**Figure 3 toxins-16-00545-f003:**
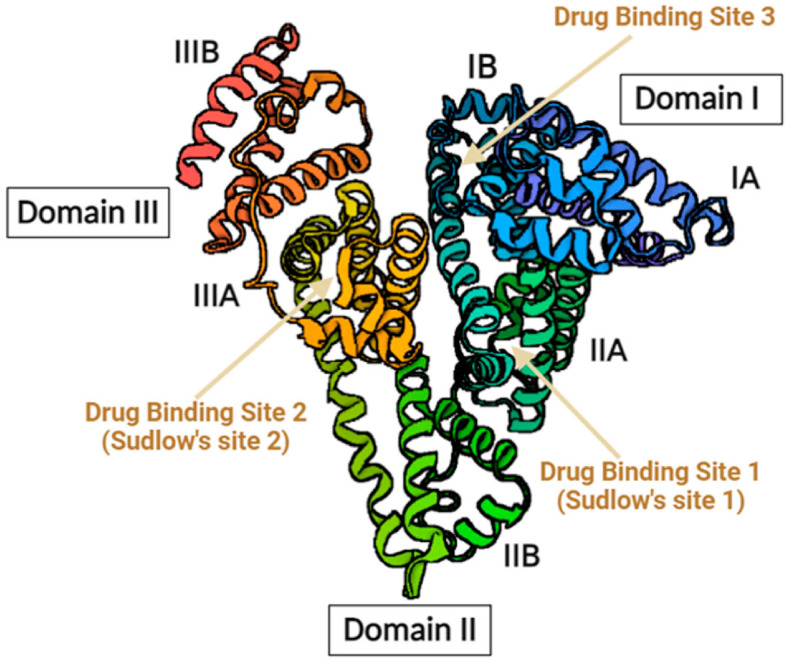
Albumin structure with three specific domains and drug-binding sites [33].

**Figure 4 toxins-16-00545-f004:**
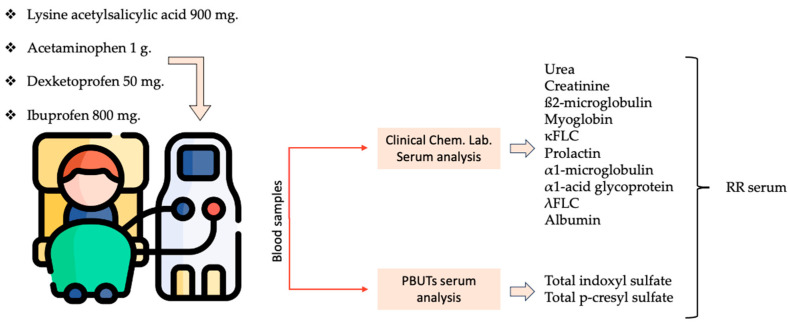
The schematic clinical set-up, sample collection, and analysis during the clinical study.

**Table 1 toxins-16-00545-t001:** Comparison of dialysis parameters in the four study sessions.

Variable	AAS	Paracetamol	Dexketoprofen	Ibuprofen
Blood processed (L)	122.7 ± 7.5	120.9 ± 7.5	120.8 ± 8.2	121.2 ± 7.1
Recirculation (%)	16 ± 5	14.7 ± 4.4	14.7 ± 4.5	16.1 ± 4.8
Real dialysis time (min)	289 ± 12	287.6 ± 11.3	288.6 ± 11.5	288.5 ± 12.4
Arterial pressure (mmHg)	−204 ± 36	−200 ± 38	−213 ± 32	−205 ± 33
Venous pressure (mmHg)	195 ± 27	199 ± 31	193 ± 31	196 ± 28
TMP (mmHg)	183 ± 38	185 ± 39	181 ± 48	182 ± 46
Initial hematocrit (%)	30 ± 5	29.1 ± 4.8	29.8 ± 5.1	29.2 ± 4.7
Final hematocrit (%)	34.7 ± 5.5	34.3 ± 5.8	34.2 ± 5.5	33.3 ± 6.1
Initial weight (kg)	69.8 ± 12.1	70.1 ± 12.3	69.9 ± 12.2	69.9 ± 12.2
Final weight (kg)	68 ± 12	68.2 ± 12.1	68.1 ± 12.1	68 ± 12
Weight gain (kg)	1.74 ± 0.83	1.93 ± 0.75	1.84 ± 0.91	1.8 ± 0.6
Replacement volume (L)	30.6 ± 4.2	31.1 ± 3.8	30.7 ± 4.7	31.4 ± 4.9

kg, kilograms; L, liters; min, minutes; mmHg, millimeters of mercury; %, percentage.

**Table 2 toxins-16-00545-t002:** Reduction ratios of a wide range of molecular weight molecules measured in all four study dialysis sessions.

Molecule	AAS	Paracetamol	Dexketoprofen	Ibuprofen
Urea RR (60 Da)	84.5 ± 3.1	85.5 ± 3.6	85.6 ± 2.9	85.3 ± 3.4
Creatinine RR (113 Da)	78.6 ± 4.1	78.9 ± 4.3	79.3 ± 3.8	78.5 ± 3.7
ß_2_-microglobulin RR (11,800 Da)	84.8 ± 3.9	85.1 ± 3.5	85.2 ± 3.1	84.8 ± 3.9
Myoglobin RR (17,200 Da)	73.2 ± 8.1	73.1 ± 7.5	72.8 ± 8.1	73.1 ± 7.6
κFLC RR (22,500 Da)	69.7 ± 6.5	69.5 ± 6.8	69.2 ± 7.3	68.8 ± 7.2
Prolactin RR (23,000 Da)	71.4 ± 9.1	70.6 ± 8.9	71 ± 9	70.2 ± 8.4
α_1_-microglobulin RR (33,000 Da)	25.8 ± 9.2	25.3 ± 11.3	27 ± 10	26.5 ± 11.1
α_1_-acid glycoprotein (41,000 Da)	13.2 ± 7.6	12.1 ± 7.6	12.9 ± 5.8	12.6 ± 6.5
λFLC RR (45,000 Da)	47.6 ± 7.9	45.6 ± 9.1	46.7 ± 11.1	46.8 ± 9.8
Albumin RR (66,000 Da)	7.5 ± 5.9	7.4 ± 4.9	8.2 ± 5.1	8.3 ± 5.2

Da, Daltons; RR, reduction ratio.

## Data Availability

The original contributions presented in this study are included in the article/Appendix A. Further inquiries can be directed to the corresponding author(s).

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
