# Peer review of "The Choice of Anti-Inflammatory Influences the Elimination of Protein-Bound Uremic Toxins"

_toxins, 2024, doi:10.3390/toxins16120545_

Round 1
Reviewer 1 Report
Comments and Suggestions for Authors
The manuscript entitled "The choice of anti-inflammatory influences the elimination of protein-bound uremic toxins" is well written and is with updated literature. Authors in the present study demonstrated that patients on chronic hemodialysis program and who report pain may benefit from intradialytic treatment with ibuprofen instead of lysine acetylsalicylic acid, paracetamol, or dexketoprofen, since in addition to improving pain, it increases the removal of protein-bound uremic toxins. The study advocate that patients with pain may benefit from treatment with ibuprofen since in addition to improving pain, it increases the removal of PBUTs.
The only limitation of the study in my opinion is that the number of subjects, to confirm it, study has to be involve more number of subjects as well as should have multicentric study.
Author Response
- The manuscript entitled "The choice of anti-inflammatory influences the elimination of protein-bound uremic toxins" is well written and is with updated literature. Authors in the present study demonstrated that patients on chronic hemodialysis program and who report pain may benefit from intradialytic treatment with ibuprofen instead of lysine acetylsalicylic acid, paracetamol, or dexketoprofen, since in addition to improving pain, it increases the removal of protein-bound uremic toxins. The study advocates that patients with pain may benefit from treatment with ibuprofen since in addition to improving pain, it increases the removal of PBUTs.
The only limitation of the study in my opinion is that the number of subjects, to confirm it, study has to be involve more number of subjects as well as should have multicentric study.
Thank you very much for your comments on our work. Our work was a proof-of-concept study, and we fully agree on the need to conduct a multicenter study using ibuprofen as a direct competitor of certain protein-bound uremic toxins. Such a study would allow for an increased patient sample size, greater statistical robustness, and subsequent research to explore its potential long-term clinical relevance.

Reviewer 2 Report
Comments and Suggestions for Authors
The manuscript's main finding is that ibuprofen administration during hemodialysis improves the elimination of IS and pCS compared to other anti-inflamatory drugs. This is clearly shown by the data presented in Figure 1.
The following points must be addressed in order to improve the quality of the manuscript:
1) A figure should be added at the end of the introduction to offer an overview of the study design. The authors could use as an example Fig. 6 in ref. 16
2) The data on the levels of IS and pCS for each patient before and after dialysis for the 4 treatments should be included in a supplementary table
3) The numbers reported in Tables 1 and 2 have many errors regarding the significant figures reported. The general rule to follow is that the standard deviation should have 2 sig. figures and the mean should stop to the same number/decimal with the st. dev.
Examples: all numbers reported for Real dialysis time, Initial weight, Final weight, and Replacement volume have wrong significant figures in Table 1
288.8+/- 12.0 should be 289+/-12
69.96+/- 12.2 should be 70+/-12
31.38+/- 4.9 should be 31.4+/-4.9
In Table 2 mistakes in α1-microglobulin, λFLC RR, and Albumin
In page 6 line 177: 70.7+/-14 should be 71+/-14
3) The discussion needs major revisions with the addition of information from the following references that clearly state the binding sites and affinities of ibuprofen, IS and pCS to Albumin.
Dehghan Niestanak, Vida, and Larry D. Unsworth. 2023. "Detailing Protein-Bound Uremic Toxin Interaction Mechanisms with Human Serum Albumin in the Pursuit of Designing Competitive Binders" International Journal of Molecular Sciences 24, 8: 7452. https://doi.org/10.3390/ijms24087452
Ghuman J, Zunszain PA, Petitpas I, Bhattacharya AA, Otagiri M, Curry S. Structural basis of the drug-binding specificity of human serum albumin. J Mol Biol. 2005;353(1):38-52. doi: 10.1016/j.jmb.2005.07.075. PMID: 16169013.
Shuhui Li, Marcello Tonelli, Larry D. Unsworth,
Indoxyl and p-cresol sulfate binding with human serum albumin,
Colloids and Surfaces A: Physicochemical and Engineering Aspects,
Volume 635, 2022, https://doi.org/10.1016/j.colsurfa.2021.128042.
The following lines of the discussion must be revised with information included in the three references mentioned above.
Lines 123-126 (Ref. 16 is not appropriate since it does not provide structural data for IS and pCS binding to Albumin)
Lines 137-138 imply that Ibuprofen binds to IS and pCS. The text should be revised with reference to Albumin binding sites and affinities for the three molecules.
Lines 139-144 refer to an allosteric mechanism, please use the structural data to clearly explain the binding sites involved.
A figure with the structure of Albumin and the binding sites of the molecules discussed would be helpful.
Comments on the Quality of English Language
In general , the quality of English is good. However, in line 151 the word "These" is not appropriate and the word "drugs" is written twice.
In line 158 "closed" must be replaced by "closely"
Author Response
2.1 A figure should be added at the end of the introduction to offer an overview of the study design. The authors could use as an example Fig. 6 in ref. 16.
Thank you for your suggestion. We have included a new figure (Figure 4) to briefly show our work’s design in the material and methods section.
2.2 The data on the levels of IS and pCS for each patient before and after dialysis for the 4 treatments should be included in a supplementary table.
We have included a supplemental table with the data from all the sessions as suggested by the reviewer.
2.3 The numbers reported in Tables 1 and 2 have many errors regarding the significant figures reported. The general rule to follow is that the standard deviation should have 2 sig. figures and the mean should stop to the same number/decimal with the st. dev.
Thank you for pointing this out. We have made the changes in both tables and in the text in order to solve these errors.
2.4 The discussion needs major revisions with the addition of information from the following references that clearly state the binding sites and affinities of ibuprofen, IS and pCS to Albumin.
- Dehghan Niestanak, Vida, and Larry D. Unsworth. 2023. "Detailing Protein-Bound Uremic Toxin Interaction Mechanisms with Human Serum Albumin in the Pursuit of Designing Competitive Binders" International Journal of Molecular Sciences 24, 8: 7452. https://doi.org/10.3390/ijms24087452
- Ghuman J, Zunszain PA, Petitpas I, Bhattacharya AA, Otagiri M, Curry S. Structural basis of the drug-binding specificity of human serum albumin. J Mol Biol. 2005;353(1):38-52. doi: 10.1016/j.jmb.2005.07.075. PMID: 16169013.
- Shuhui Li, Marcello Tonelli, Larry D. Unsworth. Indoxyl and p-cresol sulfate binding with human serum albumin. Colloids and Surfaces A: Physicochemical and Engineering Aspects. Volume 635, 2022, https://doi.org/10.1016/j.colsurfa.2021.128042.
The following lines of the discussion must be revised with information included in the three references mentioned above.
Lines 123-126 (Ref. 16 is not appropriate since it does not provide structural data for IS and pCS binding to Albumin)
Lines 137-138 imply that Ibuprofen binds to IS and pCS. The text should be revised with reference to Albumin binding sites and affinities for the three molecules.
Lines 139-144 refer to an allosteric mechanism, please use the structural data to clearly explain the binding sites involved.
Thank you very much for your valuable recommendations. You are correct in pointing out that reference [16] is not appropriate to refer to the binding sites of protein-bound uremic toxins. There was an error, and it should have been reference [26] (Watanabe, H.; Noguchi, T.; Miyamoto, Y.; Kadowaki, D.; Kotani, S.; Nakajima, M.; Miyamura, S.; Ishima, Y.; Otagiri, M.; Maruyama, T. Interaction between Two Sulfate-Conjugated Uremic Toxins, p-Cresyl Sulfate and Indoxyl Sulfate, during Binding with Human Serum Albumin. Drug Metabolism and Disposition 2012, 40, 1423 LP – 1428, doi:10.1124/dmd.112.045617]. However, the references you have provided reinforce and support the evidence regarding these albumin binding sites for protein-bound uremic toxins and other competing molecules. Therefore, we have included these references in the discussion and made some changes to the text to clarify and enhance our work.
You will observe the following modifications in lines 130-133: IS and pCS bind primarily to Sudlow site II of albumin [26-29]. This finding is supported by structural analyses highlighting their high-affinity binding patterns to this site, limiting their availability for clearance during hemodialysis [28-29].
Additionally, we have implemented the following changes in lines 146-147: Ibuprofen also showed higher binding affinity to albumin compared to IS, pCS and also to these other displacers [27,31].
Finally, to better explain the mechanism of allosteric modification, we have made the following changes to the text in lines 151-158, which we hope make the explanation clearer: Interestingly, salvianolic acids bind to Sudlow site I while IS and pCS bind predominantly to Sudlow site II [26-29], so their displacement ability is due to an allosteric mechanism instead of direct competition [34]. Moreover, free-fatty acids from IntralipidTM have competitive displacement as some have higher binding affinity to Sudlow site I, which coincides with fatty-acid (FA) binding site 7 (FA7), compared to CMPF. Others, such as oleic and linoleic acids, bind with higher affinity to Sudlow site II (FA3-FA4) compared to IS and pCS. These FA binding on albumin may change the structure of the PBUTs binding sites [34].
We trust these changes align with your suggestions and enhance the manuscript’s clarity.
2.5 A figure with the structure of Albumin and the binding sites of the molecules discussed would be helpful.
Thank you for your suggestion. In order to clarify the different binding sites of albumin, where protein-bound uremic toxins and other molecules bind, we have added to the discussion a new figure (Figure 3) that shows the tridimensional structure of albumin.
